# Self-esteem in crisis: Psychosocial adaptation and masculine identity among Chinese men with azoospermia

Fangliang Zou[1,2☉], Jue Li[3☉], Yi Fang[4☉], Jiliang Huang[5], Zikai Feng[6], Hang Shi[1,2], Yu Lan[1,2], Yang Zhang[1,2], Ruiyun Chen[1,2], Yanshan Lin[1,2*]

**1** Department of Obstetrics and Gynecology, Center for Reproductive Medicine, Guangdong Provincial Key Laboratory of Major Obstetric Diseases, Guangdong Provincial Clinical Research Center for Obstetrics and Gynecology, Guangdong-Hong Kong-Macao Greater Bay Area Higher Education Joint Laboratory of Maternal-Fetal Medicine, The Third Affiliated Hospital of Guangzhou Medical University, Guangzhou, China, **2** Guangdong Province Key Laboratory of Reproductive Medicine, The Third Affiliated Hospital of Guangzhou Medical University, Guangzhou, China, **3** Center for Nursing Research, Shantou University Medical College, Shantou, Guangdong, China, **4** School of Nursing, Shantou University Medical College, Shantou, Guangdong, China, **5** Reproductive Center, The First Affiliated Hospital of Shantou University Medical College, Shantou, Guangdong, China, **6** Department of Clinical Pharmacy, The First Affiliated Hospital of Shantou University Medical College, Shantou, Guangdong, China

☉ These authors contributed equally to this work.
* linyanshan2007@126.com

## Abstract

### Background

Azoospermia affects 1% of men and 10–20% of infertile males, yet the psychosocial mechanisms underlying self-esteem impairment remain poorly characterized. Guided by Connell's Masculinity Theory and Bury's Biographical Disruption Framework, this mixed methods study examined self-esteem experiences among Chinese men with azoospermia.

### Methods

An explanatory sequential design was employed. Phase 1 surveyed 216 men using Rosenberg's Self-Esteem Scale, with multiple regression identifying predictors. Phase 2 involved semi-structured interviews with 16 purposively sampled participants, analyzed through thematic analysis. Integration of quantitative and qualitative findings provided comprehensive interpretation.

### Results

Quantitative analysis revealed moderate self-esteem overall (mean = 30.18 ± 3.99), with 10.19% exhibiting low self-esteem. Significant predictors included monthly income ($\beta = 0.210$, $p < 0.001$), family harmony ($\beta = 0.141$, $p = 0.028$), and attitudes toward childbearing discussions ($\beta = 0.159$, $p = 0.014$). Qualitative findings identified

**Data availability statement:** Quantitative survey data (n=216) are publicly available from the Zenodo repository at https://doi.org/10.5281/zenodo.17641014. Qualitative interview data cannot be shared publicly due to ethical restrictions: transcripts contain sensitive personal narratives about stigmatized topics that risk participant identification, and the Institutional Review Board (No. 2019-066) prohibited public deposition as participants consented to research use only. Qualified researchers may request access to qualitative data from the Ethics Committee (gysygcpiec@126.com, Tel: 020-81292726) or corresponding author Yanshan Lin (linyanshan2007@126.com) following ethics review.

**Funding:** The author(s) received no specific funding for this work.

**Competing interests:** The authors have declared that no competing interests exist.

five themes, including economic burden, family dynamics, social stigma, treatment uncertainty, and cumulative psychological impacts. Integration demonstrated financial capacity's dual role as both practical enabler and symbolic compensation for perceived masculine failure.

## Conclusions

Self-esteem in azoospermic men is shaped by interconnected economic, familial, and social factors. Clinical interventions should integrate financial counseling, family-based emotional support, and culturally tailored stigma reduction, highlighting the value of mixed methods in understanding male infertility psychosocial dimensions.

---

## Introduction

Infertility affects approximately 8–12% of couples worldwide, with male factors contributing to nearly half of these cases [1]. Among the various causes of male infertility, azoospermia, defined as the complete absence of sperm in the ejaculate, represents one of the most severe forms, affecting approximately 1% of all men and 10–20% of infertile males [2,3]. Notably, over the past three decades, the prevalence of azoospermia has increased by nearly 25%, alongside a significant 50% decline in overall sperm quality, posing a critical challenge to male reproductive health [4]. Unlike other forms of male infertility, azoospermia is characterized by irreversible spermatogenic failure in most cases, leaving affected men with limited treatment options [5,6].

Azoospermia imposes significant psychological and emotional burdens on affected individuals. Men diagnosed with this condition often experience heightened levels of anxiety, depression, guilt, and diminished self-esteem [7,8]. Although research suggests that men and women experience comparable levels of psychological distress in infertility, men tend to adopt avoidant coping mechanisms, which may lead to emotional struggles being overlooked or underestimated [5,9]. Critically, in collectivist societies like China where Confucian values emphasize patrilineal continuity, male infertility remains profoundly stigmatized, with fertility directly linked to cultural perceptions of masculinity, virility, and social status [10]. The perception of infertility as a failure to fulfill societal expectations can lead to marital stress, social exclusion, and self-stigmatization, further compounding the psychological distress faced by azoospermic men [11].

Additionally, the medical management of azoospermia often involves invasive surgical procedures such as testicular sperm extraction (TESE) or microdissection to retrieve viable sperm [12]. Despite technological advances like micro-TESE offer improved sperm retrieval rates [13], treatment outcomes remain uncertain. Even in cases where sperm retrieval is successful, fertilization and clinical pregnancy rates are significantly lower than natural conception [14]. The persistent uncertainty of treatment outcomes, coupled with the irreversible nature of azoospermia in many cases, exacerbates feelings of helplessness and self-worth deterioration [15].

Existing research suggests that azoospermic men report significantly lower self-esteem compared to other infertile men [8]. Self-esteem, defined as the subjective evaluation of one's worth and sense of being valued [16], plays a pivotal role in psychological well-being. For men in patriarchal societies, the inability to father biological children often triggers a "dual failure": compromising both the biological progenitor and economic provider roles central to hegemonic masculinity [17].

Theoretical framing is essential to decode this complexity. Connell's Masculinity Theory [18] posits that societal expectations of male dominance and fertility shape identity construction. When biological reproduction fails, men may disproportionately invest in economic success as compensatory masculinity validation—a mechanism requiring empirical verification. Concurrently, Bury's Biographical Disruption Framework [19] elucidates how chronic illness triggers identity renegotiation, yet prior studies neglect how azoospermic men navigate this through alternative fatherhood pathways (e.g., sperm donation acceptance) versus social withdrawal.

Notably, while quantitative studies have documented self-esteem deficits in azoospermic men [8,20], their lived experiences navigating cultural expectations remain underexplored in China's context of rapid urbanization clashing with traditional patrilineal values [21]. This intersection creates unique psychological challenges that demand investigation. Therefore, this study employs a mixed-methods approach to quantify self-esteem predictors, explore subjective experiences, and investigate how cultural masculinity scripts shape psychological adaptation. By dissecting economic duality in treatment access and masculine identity compensation, we pioneer identity reconstruction pathways beyond biological fatherhood. Ultimately, this work delivers culturally-calibrated frameworks to dismantle infertility stigma across societal spheres, transforming Confucian burdens into resilience narratives. Its seminal contribution transcends victimhood documentation to empower men navigating the irreconcilable duality of ancestral duties and biological realities, repositioning them as architects of redefined masculinity.

## Methods

### Study design

This study adopted an explanatory sequential mixed-methods design guided by Connell's Masculinity Theory [18] and Bury's Biographical Disruption Framework [19]. In this approach, the initial quantitative phase comprised a cross-sectional survey using structured questionnaires to assess participant characteristics and self-esteem scores. Subsequently, the qualitative phase built upon these quantitative findings through semi-structured interviews, aimed at exploring in-depth the lived experiences of self-esteem among men with azoospermia. Throughout the research process, quantitative and qualitative data were collected and analyzed independently. Finally, the datasets were integrated, with qualitative insights specifically utilized to explain and comprehensively contextualize the interpretations derived from the quantitative phase.

### Phase 1: Quantitative survey

**Participants and setting.** Participants were recruited through convenience sampling from male patients diagnosed with azoospermia at the Center for Reproductive Medicine at the Third Affiliated Hospital of Guangzhou Medical University from August to December 2023. After receiving a brief introduction to the study's purpose and procedures, participants provided informed consent. Eligible participants then completed paper questionnaires administered face-to-face by researchers. Inclusion criteria required participants to: (1) have a confirmed azoospermia diagnosis via semen analysis according to WHO criteria [22]; (2) be able to read, understand, and complete the questionnaires; (3) demonstrate emotional stability, strong communication skills, and clear speech; and (4) voluntarily participate and cooperate with the study. Exclusion criteria were: (1) patients with poor semen analysis results attributable to other organic diseases or medications; (2) patients withdrawing midway through the study; (3) patients with other major comorbidities or psychiatric disorders; (4)patients with severe sexual dysfunction, such as erectile dysfunction, low libido and premature ejaculation. In compliance with China's Asstied Reproductive Technology(ART) regulations, all participants were legally married, as ART services are restricted to married couples with documented infertility [23,24].

Sample size calculation was based on available prevalence data from the literature. Given the absence of large-scale studies examining self-esteem levels specifically among Chinese men with azoospermia, we referenced a previous study on Polish transgender individuals which reported a low self-esteem prevalence of 11% [25]. Using this conservative estimate and applying the standard sample size formula for cross-sectional studies (95% confidence interval, 5% margin of error), a minimum sample size of 168 participants was required, accounting for a 10% non-response rate.

Formula:

$$n = \frac{Z^2 a/2 \, P(1-P)}{\delta^2}$$

**Instruments.** Quantitative data were collected using a questionnaire comprising the following sections: basic demographic information (ethnicity, religious belief, education level, occupation, long-term residence, family monthly income per capita), disease-related information (cohabitation duration, infertility duration, treatment duration, and infertility causes), and the Self-Esteem Scale (SES). The SES, developed by Morris Rosenberg [26], is a 10-item self-evaluation instrument. Responses are recorded on a 4-point Likert scale (1 = Very consistent to 4 = Very inconsistent). Total scores range from 10 to 40, with higher scores indicating greater self-esteem. Scores were categorized as follows: ≤ 25 (low self-esteem), 26–32 (moderate self-esteem), and ≥33 (high self-esteem). This scale has been widely used and validated in the Chinese population [27]. In the present study, the scale demonstrated good internal consistency, with a Cronbach's α coefficient of 0.830.

**Data analysis.** Data were recorded in Excel and analyzed using SPSS (version 26.0). Continuous variables were assessed for normality via Q-Q and P-P plots. Descriptive statistics (frequencies, percentages, means, standard deviations) summarized participant characteristics. Group comparisons used independent t-tests or ANOVA. Multiple linear regression identified associations between predictors and self-esteem scores, with statistical significance set at $p < 0.05$.

### Phase 2: qualitative survey

**Participants and sample size.** The inclusion and exclusion criteria for the qualitative phase were identical to those used in the quantitative phase. Upon completing the questionnaire, participants were asked whether they were willing to participate in an interview to further explore their self-esteem experiences related to living with azoospermia. A total of 34 participants provided contact information for follow-up. Interview invitations were randomly distributed to eligible participants from the quantitative study who had expressed interest in being interviewed. Prior to scheduling, participants were contacted by telephone to arrange interview sessions at their convenience. Written informed consent was obtained before each interview commenced. Interviews were conducted until data saturation was achieved, defined as the point at which no new themes or insights emerged from additional interviews [28]. Data saturation was reached after 16 interviews.

**Data collection.** Semi-structured interviews were conducted between May 5 and July 11, 2024, in a private consultation room at the Center for Reproductive Medicine, Third Affiliated Hospital of Guangzhou Medical University. Each interview lasted approximately 15–30 minutes. All interviews were conducted in Mandarin by a single researcher with extensive qualitative research experience and specialized expertise in reproductive health. The researcher had received formal training in qualitative interviewing techniques and adopted an interpretative phenomenological approach [29], which emphasizes the researcher's active engagement in both experiencing and analyzing the phenomenon under study.

An interview guide was developed based on emerging themes from Phase 1 [30], focusing on participants' experiences and emotional journeys following an azoospermia diagnosis. Prior to each interview, written informed consent

was obtained, including explicit permission for audio recording. With participants' approval, all interviews were digitally recorded. The researcher also maintained detailed observational notes, documenting non-verbal cues such as tone shifts, pauses, gaze direction, hesitations, prolonged silences, and other behavioral indicators. To ensure data accuracy, the researcher summarized key points at the end of each session and verified them with participants.

Audio recordings were transcribed verbatim within 24 hours by the research team. A second researcher independently reviewed the recordings and cross-checked transcripts to ensure precision and completeness. The interview guide was systematically developed based on Phase 1 quantitative findings and refined through consultations with two reproductive medicine specialists and one psychologist. Prior to the main study, the guide was pilot-tested with two eligible participants not included in the final sample.

Data collection encompassed both demographic information (age, marital age, infertility history, education level, time since diagnosis, and treatment duration) and semi-structured interviews. The interview outline explored: 1) initial emotional reactions to the azoospermia diagnosis; 2) family responses and their impact on emotional well-being and self-esteem; 3) perceptions of financial burdens from treatment and their influence on self-worth; 4) social interactions involving fertility discussions and their psychological effects; and 5) emotional resilience and self-esteem changes throughout the treatment journey. Each interview concluded with an open-ended invitation: *"Is there anything else you'd like to share about your experience? Do you have any questions for me?"* This allowed participants to contribute additional insights freely.

**Data analysis.** Interview transcripts were produced verbatim within 24 hours of each session, with written observational notes carefully reviewed to ensure transcription accuracy. Data management and analysis were conducted using NVivo 11.0 qualitative research software. Following Braun & Clarke's thematic analysis approach [31], two researchers independently analyzed the data through a six-phase process: 1) immersive reading of transcripts to achieve comprehensive familiarity with the data; 2) systematic coding of meaningful data units; 3) organization of codes into potential themes; 4) critical theme review; 5) theme definition and naming; and 6) report generation. A third researcher compared the independently derived coding schemes and thematic structures, with any discrepancies resolved through group discussion to reach consensus on the final analytical framework.

**Rigor.** Multiple strategies were implemented to ensure methodological rigor. First, the research team comprised members with specialized training in qualitative methods, extensive interviewing experience, and professional expertise in infertility-related research. Second, team members represented diverse professional backgrounds and maintained continuous collaborative discussions throughout the research process. During data analysis, all team members engaged in iterative data review and collective interpretation until reaching consensus. Additionally, the inclusion of bilingual researchers facilitated accurate translation of textual materials. These measures collectively enhanced the credibility and trustworthiness of both the research process and findings.

**Integration of quantitative and qualitative data.** The study employed an integrative approach to strengthen methodological rigor by examining convergence, divergence, and complementarity between quantitative and qualitative findings [32]. Qualitative results were systematically used to contextualize and interpret the quantitative data. The integration process involved: 1) independent analysis and presentation of each dataset; 2) synthesis of findings through comparative analysis; and 3) development of comprehensive interpretations that elucidate the self-esteem experiences of men with azoospermia. The research team conducted thorough investigations to account for any observed discrepancies between datasets. By prioritizing the qualitative phase, we were able to provide nuanced explanations for the quantitative results obtained in Phase 1.

## Ethical considerations

This study received ethical approval from the Institutional Review Board of the Third Affiliated Hospital of Guangzhou Medical University (Approval No: 2019−066) and adhered to the principles of the Declaration of Helsinki. All participants

provided written informed consent after receiving complete information about the study procedures, with explicit assurance of their right to withdraw at any time without consequence. Data confidentiality was rigorously maintained through de-identification procedures and secure storage protocols.

## Results

### Quantitative phase

**Characteristics of the sample.** The quantitative phase included 216 men diagnosed with azoospermia. As presented in Table 1, participants had a mean age of 33.06 ± 5.37 years and mean marriage duration of 5.78 ± 4.07 years. Additional reproductive characteristics included average cohabitation duration 6.12 ± 4.04 years, infertility duration 4.64 ± 3.08 years, treatment duration 2.66 ± 2.14 years, and the gap time from discovering infertility to starting treatment was 1.98 ± 2.46 years. It should be noted that some participants had initiated fertility treatment prior to receiving a definitive azoospermia diagnosis. Self-esteem scores demonstrated a normal distribution with a mean of 30.18 ± 3.99, reflecting moderate overall self-esteem levels. Notably, 10.19% of participants scored in the low self-esteem range.

**Factors associated with self-esteem.** Univariate analysis revealed several significant factors associated with self-esteem scores (Table 1). Educational attainment (F = 4.550, $p = 0.004$), geographic residence (F = 4.114, $p = 0.018$), and household income per capita (F = 7.504, $p < 0.001$) emerged as significant socioeconomic predictors. Regarding psychosocial factors, family relationship harmony (t = −2.588, $p = 0.010$) and attitudes toward childbearing (F = 3.401, $p = 0.019$) showed statistically significant associations with self-esteem levels.

The self-esteem scores of men with azoospermia were analyzed as the dependent variable in this study. Based on preliminary univariate analysis, demographic variables demonstrating statistically significant associations ($p < 0.05$) were selected as independent variables for subsequent multivariate linear regression modeling. Diagnostic tests confirmed the absence of multicollinearity among predictor variables (Table 2). The final regression model incorporated four significant predictors: education level, average monthly household income, family relationship harmony, and attitudes toward childbearing. Each of these variables made statistically significant contributions ($p < 0.05$) to explaining variance in self-esteem scores.

### Qualitative phase

**Characteristics of the sample.** The qualitative phase included 16 men with azoospermia who participated in semi-structured interviews. Participants ranged in age from 26 to 49 years (mean age 33.25 ± 5.46 years), with infertility durations spanning 2–11 years and treatment durations ranging from 1 to 8 years. Additional demographic and professional characteristics of the study participants are detailed in Table 3.

**Findings from thematic synthesis.** Five overarching themes emerged from the qualitative data, providing nuanced explanations for the quantitative findings: 1) economic constraints shaping treatment experiences; 2) family dynamics influencing self-perception; 3) social stigma and communication challenges; 4) treatment-related uncertainty affecting psychological resilience; and 5) cumulative psychological impacts. Each theme comprised multiple subthemes, as detailed in the following sections.

**Theme 1: Economic constraints shaping treatment experience.** *Subtheme 1.1: Financial burden exacerbating self-esteem issues*: Participants consistently reported that the substantial financial demands of azoospermia treatment significantly undermined their self-esteem. The economic strain frequently manifested as feelings of guilt, self-reproach, and perceived inadequacy.

*"Each unsuccessful treatment cycle feels like wasted money, intensifying my guilt and sense of worthlessness. The financial stress becomes overwhelming." (N7)*

**Table 1. Demographic characteristics and univariate analysis of the participants from the quantitative phase (n=216).**

| Variable | Category | n (%) | SES score (Mean±SD) | t/F value | p value |
|---|---|---|---|---|---|
| Age(year) | ≤30 | 88 (40.7) | 30.03±4.25 | 1.057 | *0.368* |
| | 31~35 | 58 (26.9) | 30.47±3.90 | | |
| | 36~40 | 49 (22.7) | 29.61±3.18 | | |
| | ≥41 | 21 (9.7) | 31.33±4.70 | | |
| Religious beliefs | No | 195 (90.3) | 30.24±3.93 | 0.679 | 0.498 |
| | Yes | 21 (9.7) | 29.62±4.53 | | |
| Education level | Junior high school and below | 54 (25.0) | 28.96±2.98 | 4.550 | 0.004 |
| | High school or junior college | 60 (27.8) | 29.60±3.48 | | |
| | College/Bachelor's Degree | 95 (44.0) | 31.09±4.43 | | |
| | Master's degree or above | 7 (3.2) | 32.14±5.43 | | |
| Residence | Big cities | 79 (36.6) | 31.16±0.46 | | |
| | Second tier cities and counties | 72 (33.3) | 29.82±4.02 | 4.114 | 0.018 |
| | Countryside | 65 (30.1) | 29.38±3.61 | | |
| Average monthly household income per person (**CNY**) | ≤3000 | 25 (11.6) | 28.40±2.84 | 7.504 | <0.001 |
| | 3001-5000 | 79 (36.6) | 29.39±3.85 | | |
| | 5001-10000 | 68 (31.5) | 30.38±4.02 | | |
| | ≥10001 | 44 (20.4) | 32.30±3.90 | | |
| Marital status | First marriage | 199 (92.1) | 30.17±3.97 | −0.185 | 0.853 |
| | Remarriage | 17 (7.9) | 30.35±4.24 | | |
| Family living arrangement | Couples living apart | 13 (6.0) | 29.31±3.92 | 1.072 | 0.344 |
| | Living with spouse | 124 (57.4) | 30.51±3.89 | | |
| | Living with parents | 79 (36.6) | 29.81±4.14 | | |
| Family relationship harmony | No | 23 (10.6) | 28.17±3.26 | −2.588 | 0.010 |
| | Yes | 193 (89.4) | 30.42±4.01 | | |
| Parental status | No (No children) | 176 (81.5) | 30.22±4.03 | 0.317 | 0.752 |
| | Yes (Have children) | 40 (18.5) | 30.00±3.82 | | |
| Urgency of having children | Immediate desire for children | 125 (57.9) | 30.20±3.92 | 0.072 | 0.930 |
| | Desire to have children soon | 86 (39.8) | 30.12±3.94 | | |
| | Can wait a little longer | 5 (2.3) | 30.80±6.83 | | |
| Wife's urgency to have children | Very urgent | 193 (89.4) | 30.19±4.05 | 0.119 | 0.905 |
| | Not urgent | 23 (10.6) | 30.09±3.53 | | |
| Attitude toward childbearing questions | Refuse to answer | 2 (0.9) | 27.00±4.24 | 3.401 | 0.019 |
| | Change the topic | 13 (6.0) | 27.08±5.50 | | |
| | Give a reserved response | 85 (39.4) | 30.35±3.60 | | |
| | Answer honestly | 116 (53.7) | 30.46±3.95 | | |
| Cause of infertility | Male factor | 117 (54.2) | 29.79±4.03 | −1.551 | 0.122 |
| | Both partner factors | 99 (45.8) | 30.64±3.91 | | |
| Infertility time(year) | ≤3 | 102 (47.3) | 30.56±4.11 | 2.503 | *0.060* |
| | 4~6 | 61 (28.2) | 30.64±3.85 | | |
| | 7~9 | 32 (14.8) | 28.66±3.96 | | |
| | ≥10 | 21 (9.7) | 29.33±3.31 | | |
| Treatment time(year) | ≤1 | 83 (38.4) | 30.40±3.99 | 1.931 | *0.126* |
| | 2~3 | 79 (36.6) | 30.54±4.04 | | |
| | 4~5 | 34 (15.7) | 29.94±4.08 | | |
| | ≥6 | 20 (9.3) | 30.18±3.99 | | |

*(Continued)*

**Table 1.** (Continued)

| Variable | Category | n (%) | SES score (Mean ± SD) | t/F value | p value |
|---|---|---|---|---|---|
| Number of ART treatment cycles | 0 | 148 (68.5) | 30.38 ± 3.98 | 1.136 | 0.336 |
| | 1 | 29 (13.4) | 30.38 ± 3.74 | | |
| | 2 | 27 (12.5) | 29.70 ± 4.24 | | |
| | ≥3 | 12 (5.6) | 28.33 ± 3.94 | | |
| Typer of azoospermia | Non-obstructive azoospermia (NOA) | 101(46.8) | 30.55 ± 3.80 | 1.67 | 0.197 |
| | Obstructive azoospermia (OA) | 115(53.2) | 29.85 ± 4.13 | | |

SD, standard deviation; t, independent samples t-test statistic; F, analysis of variance (ANOVA) F-value; p, significance level; n (%), frequency and percentage. Based on 2024 average exchange rates (1 CNY ≈ 0.14 USD ≈ 0.13 EUR), the income brackets correspond approximately to ≤3000 CNY (≤\$420/€390), 3001–5000 CNY (\$420–700/€390–650), 5001–10000 CNY (\$700–1400/€650–1300), and ≥10001 CNY (≥\$1400/€1300).

**Table 2.** Multiple linear regression analysis of factors influencing self-esteem.

| Independent variable | B | SE | β | t | p Value | LLCI | ULCI | Tol | VIF |
|---|---|---|---|---|---|---|---|---|---|
| Constant term | 19.419 | 2.196 | — | 8.843 | <0.001 | 15.090 | 23.749 | — | — |
| Education level | 0.725 | 0.324 | 0.158 | 2.236 | 0.026 | 0.086 | 1.364 | 0.802 | 1.246 |
| Average monthly household income per person | 0.893 | 0.302 | 0.210 | 2.958 | 0.003 | 0.298 | 1.488 | 0.796 | 1.256 |
| Family relationship harmony | 1.821 | 0.825 | 0.141 | 2.208 | 0.028 | 0.195 | 3.447 | 0.983 | 1.017 |
| Attitude toward childbearing questions | 0.969 | 0.391 | 0.159 | 2.479 | 0.014 | 0.198 | 1.739 | 0.982 | 1.018 |

$R^2 = 0.151$, Adjusted $R^2 = 0.135$, F = 9.370, $p < 0.001$. B, unstandardized coefficient; SE, standard error; β, beta, standardized coefficient; CI, confidence interval; Tol, tolerance; VIF, variance inflation factor; LLCI, lower level of confidence interval; ULCI, upper level of confidence interval.

*"Observing our savings diminish is profoundly distressing. Assuming responsibility for this financial depletion substantially erodes my self-confidence."* (N14)

*"I constantly worry about the economic consequences for my family. Each payment reinforces my perception of being a burden and further diminishes my self-esteem."* (N2)

**Subtheme 1.2: Economic capacity and masculine identity**: A prominent pattern emerged linking participants' financial resources to their conceptions of masculinity, particularly when confronting fertility challenges.

*"As the primary provider, I should be capable of financing treatment, but each medical bill leaves me feeling doubly powerless."* (N1)

*"My earnings were intended to support my family's future, yet they're now consumed by treatment costs. This reality makes me question my masculine identity."* (N11)

*"When biological fatherhood proves impossible, financial provision becomes the remaining marker of masculinity. When treatment expenses jeopardize even this, the sense of inadequacy becomes complete."* (N3)

**Theme 2: Family dynamics and self-perception.** **Subtheme 2.1: Intergenerational expectations and emotional burden**: The cultural imperative for patrilineal continuity in Chinese families emerged as a significant source of psychological distress, with participants reporting profound feelings of guilt and diminished self-worth related to parental expectations.

**Table 3. Characteristics of the participants from the qualitative phase (N = 16).**

| No. | Age | Education | Residence | Monthly income per person (CNY) | Reproductive status | Infertility factors | Years of infertility | Years of treatment | Typer of azoospermia |
|---|---|---|---|---|---|---|---|---|---|
| N1 | 35 | Junior high school | Big cities | 5001~10000 | No children | Male | 11 | 1 | NOA |
| N2 | 28 | High school | Small towns | 5001~10000 | No children | Both | 2 | 0.5 | OA |
| N3 | 33 | Vocational School | County | 3001~5000 | No children | Male | 8 | 8 | NOA |
| N4 | 36 | Junior college | Big cities | ≥10001 | Has children | Male | 1.5 | 1 | OA |
| N5 | 29 | Undergraduate | Big cities | ≥10001 | No children | Male | 3 | 2 | NOA |
| N6 | 49 | Junior college | County | 5001~10000 | Has children | Both | 2 | 1 | OA |
| N7 | 35 | Junior college | Big cities | 3001~5000 | No children | Both | 3 | 2.5 | NOA |
| N8 | 26 | Junior high school | Rural | ≤3000 | No children | Male | 4.5 | 3 | NOA |
| N9 | 29 | High school | County | 5001~10000 | No children | Both | 3 | 1 | NOA |
| N10 | 36 | Undergraduate | Big cities | ≥10001 | Has children | Male | 1.5 | 1 | OA |
| N11 | 37 | Junior high school | Big cities | ≥10001 | No children | Male | 11 | 1 | OA |
| N12 | 33 | Vocational School | County | 3001~5000 | No children | Male | 8 | 8 | NOA |
| N13 | 35 | Junior high school | County | ≤3000 | Has children | Male | 2 | 2 | OA |
| N14 | 33 | High school | County | ≥10001 | No children | Both | 3 | 0.5 | OA |
| N15 | 31 | High school | County | 3001~5000 | No children | Both | 6 | 3 | OA |
| N16 | 27 | Vocational School | County | 3001~5000 | No children | Both | 5 | 3 | NOA |

Based on 2024 average exchange rates (1 CNY ≈ 0.14 USD ≈ 0.13 EUR), the income brackets correspond approximately to ≤3000CNY (≤$420/€390), 3001–5000 CNY ($420–700/€390–650), 5001–10000 CNY ($700–1400/€650–1300), and ≥10001 CNY (≥$1400/€1300). Reproductive status indicates whether participants had biological children prior to azoospermia diagnosis. "No children" represents primary azoospermia (men who have never fathered children), while "Has children" represents secondary azoospermia (men who previously fathered biological children through natural conception before developing azoospermia).

*"My aging father longs to hold his grandson, and the family has invested both emotionally and financially in my treatment. While it's my biological issue, the burden extends to my parents..." (N2)*

*"Though my parents avoid direct comments, I'm acutely aware of their unspoken expectations regarding grandchildren. Despite their consideration, I'm consumed by shame and feel unable to meet familial obligations." (N3)*

*"Seeing my father-in-law interact with his friends' grandchildren while regarding me differently, that silent disapproval is psychologically devastating." (N8)*

***Subtheme 2.2: Marital support systems:*** The quality of spousal relationships played a pivotal role in moderating self-esteem impacts, with supportive partnerships providing emotional buffers while strained dynamics amplified distress.

*"My wife is not the issue, it's mainly me. On one hand, I do feel sorry for her. On the other hand, knowing this, I dare not say much to her for fear that she might look down on me." (N6)*

*"My wife really loves children. Even though she found out that I have no sperm, she didn't complain and has been facing it with me. I still feel very guilty towards her. If she stays with me through this, I will repay her with all my heart." (N15)*

*"My wife's continuous support has been crucial. Without her, coping emotionally would have been nearly impossible. She helps me maintain some confidence."* (N5)

**Theme 3: Social stigma and communication challenges.** *Subtheme 3.1: Strategic disclosure management*: Participants consistently described infertility as a stigmatized condition requiring careful information control to preserve social standing and self-worth.

*"This remains strictly our private marital matter. We're determined to handle it discreetly to maintain our dignity."* (N5)

*"Having previously fathered a child makes this situation particularly humiliating. I'm adamant about keeping this private at my stage of life."* (N6)

*"Seeking treatment in an unfamiliar location provides necessary anonymity. The prospect of community gossip about my condition is terrifying!"* (N7)

*Subtheme 3.2: Social withdrawal patterns*: Avoidance emerged as a predominant but ultimately detrimental coping mechanism, creating cyclical patterns of isolation and diminished self-esteem.

*"I actively avoid social events where fertility might be discussed. Constantly dodging questions hurts my self-worth and deepens my sense of isolation."* (N8)

*"Isolation feels safer than facing questions about fertility, but in the end, it leaves me feeling lonelier and further reduces my self-esteem."* (N12)

*"When people used to ask me when I'm having kids, I'd snap back at them, saying it's none of their business. Now I just avoid gatherings entirely."* (N8)

*"I usually avoid this topic with colleagues and friends, and they know I don't like to talk about it. Over time, they've stopped inviting me to family-centered events altogether."* (N14)

*Subtheme 3.3: Emotional constriction*: Participants reported significant difficulties in emotional expression regarding their condition, with suppressed feelings exacerbating psychological distress.

*"Family celebrations have become minefields of discomfort. Even without verbal confirmation, my childless status at this age feels like a glaring deficiency that lowers my social standing."* (N11)

*"I reject the 'sick' label and I'm physically healthy. This conceptual disconnect makes it impossible to seek appropriate emotional support when needed most."* (N1)

**Theme 4: Treatment uncertainty and psychological adaptation.** *Subtheme 4.1: Emotional volatility from prolonged uncertainty*: The unpredictable nature of treatment outcomes created profound psychological distress, undermining participants' sense of agency and self-worth over time.

*"Years of unsuccessful attempts have left me emotionally depleted, constantly searching for solutions but ultimately feeling like a complete failure."* (N3)

*"As a physically healthy individual, receiving no clear explanation for my azoospermia is profoundly destabilizing. Each unsuccessful intervention, from biopsies to medications, compounds my sense of helplessness."* (N13)

*"The not knowing is worse than a definitive negative. Each treatment brings hope that quickly transforms into disappointment, creating emotional whiplash."* (N16)

*Subtheme 4.2: Clinical encounters as emotional modifiers*: The nature of medical interactions significantly mediated participants' coping mechanisms and self-perception.

*"The andrology doctor really understands our situation and listens to me patiently. This condition's complexity makes it nearly impossible to discuss meaningfully outside clinical settings!" (N12)*

*"Some physicians treated my condition so clinically that it made me feel like a failed specimen rather than a person, exacerbating my shame. Others recognized the human dimension, helping preserve my sense of dignity amidst medical challenges." (N9)*

**Theme 5: Progressive psychosocial deterioration.** *Subtheme 5.1: Personality restructuring*: Participants reported fundamental changes in temperament and worldview resulting from chronic treatment stress and social strain.

*"I've been watching this for years. My temper has been getting worse and worse. I'm easily annoyed and upset, and I doubt whether I'm really competent!" (N1)*

*"Sometimes I feel like I don't get along with anyone. It feels like they're all against me, and they just don't understand." (N12)*

*"The cheerful, confident person I was before diagnosis has gradually disappeared. I've become more pessimistic and defensive in all areas of life." (N8)*

*Subtheme 5.2: Vocational and social disintegration*: The cumulative burden manifested in tangible professional consequences and social withdrawal.

*"Because of this, I had arguments with others, lost my job, and now I don't even go out anymore. I don't want to get myself upset and upset others." (N8)*

*"Because I've been taking so many sick days, and it's not easy to explain to my workplace that I'm seeing a doctor for this issue, I've now dedicated all my time to dealing with it. It's really made me depressed." (N16)*

### Integration of quantitative and qualitative findings

The integration of quantitative and qualitative findings provided a comprehensive understanding of self-esteem experiences among men with azoospermia. Following the explanatory sequential design, the qualitative data offered deeper insights into the statistical relationships identified in the quantitative phase. Table 4 presents this integration as a joint display showing areas of convergence, complementarity, and divergence. The mixed methods findings reveal a complex, interconnected web of factors influencing self-esteem among men with azoospermia (Fig 1). Grounded in Connell's Masculinity Theory and Bury's Biographical Disruption Framework, this conceptual model illustrates how cultural expectations of masculinity create a context where economic, familial, and social factors interact to shape psychological outcomes.

### Discussion

This explanatory sequential mixed-methods study offers novel insights into the multifaceted nature of self-esteem among men with azoospermia. Quantitatively, we identified significant associations between lower self-esteem and three key factors: reduced household income, poorer family relationship quality, and avoidant responses to fertility-related discussions (all $p < 0.05$). Qualitative data enriched these findings by elucidating the underlying mechanisms (financial strain, intergenerational expectations, social stigma, and treatment fatigue) collectively shaped participants' self-perceptions through culturally constructed masculine ideals. The integrated analysis underscores how azoospermia imposes a

 

**Table 4. Integration of quantitative and qualitative findings.**

| Quantitative Finding | Qualitative Explanation | Synthesis |
|---|---|---|
| Monthly household income was significantly associated with self-esteem (β = 0.210, $p < 0.001$) | Thematic analysis revealed two distinct yet interrelated economic dimensions: 1) treatment-related financial strain (*"Every unsuccessful treatment feels like throwing away money"* - N7); 2) masculinity compensation through economic provision (*"When you can't father a child naturally, you at least want to demonstrate value through financial provision"* - N3). | The quantitative association between income and self-esteem manifests through dual pathways: 1) directly by enabling treatment access; 2) symbolically by serving as compensatory masculinity validation, particularly salient in contexts of reproductive failure. |
| Family relationship harmony was significantly associated with self-esteem (β = 0.141, $p = 0.028$) | Qualitative narratives exposed paradoxical family dynamics: 1) cultural expectations generating intergenerational guilt (*"It's my problem, but my parents have to worry and spend money for me..."* - N2); 2) marital support buffering distress (*"My wife's continuous support has been crucial"* - N5). | While harmonious relationships quantitatively predict better self-esteem, qualitative data reveal this protective effect operates alongside significant cultural pressures. The net psychological impact reflects a balance between these opposing forces within familial contexts. |
| Attitudes toward fertility questions were significantly associated with self-esteem (β = 0.159, $p = 0.014$) | Participants described sophisticated stigma management strategies: 1) information control (*"This remains strictly our private marital matter"* - N5); 2) social withdrawal (*"I actively avoid social events where fertility might be discussed"* - N8); 3) emotional constriction (*"Family celebrations have become minefields of discomfort"* - N11). | Quantitative measures of fertility-related attitudes capture surface manifestations of deeper stigma processes revealed qualitatively. Avoidance behaviors represent both protective mechanisms and maladaptive coping strategies that ultimately reinforce low self-esteem. |
| Education level was statistically significant in regression models (β = 0.158, $p = 0.026$) | While education provided cognitive resources (*"I research everything I can about this condition...it gives me some sense of control"* - N5), its buffering capacity was limited against: 1) treatment fatigue (*"Because I've been taking so many sick days... I've now dedicated all my time to dealing with it"* - N16); 2) identity threats (*"The cheerful, confident person I was before diagnosis has gradually disappeared"* - N8). | Education's protective effect operates primarily through enhanced health literacy and coping skills, but proves insufficient to fully counteract the cumulative psychological impacts of prolonged treatment and identity disruption. |
| No significant association between treatment duration and self-esteem | Prolonged treatment uncertainty and repeated failures induced significant emotional distress: *"After years of treatment, I feel like a complete failure. The constant cycle of hope and disappointment has left me emotionally numb"* (N3). | Methodological divergence: While quantitative analysis showed no direct correlation, qualitative findings revealed profound psychological impacts of extended treatment duration, indicating that standard measures may not fully capture the cumulative emotional toll of chronic treatment experiences. |

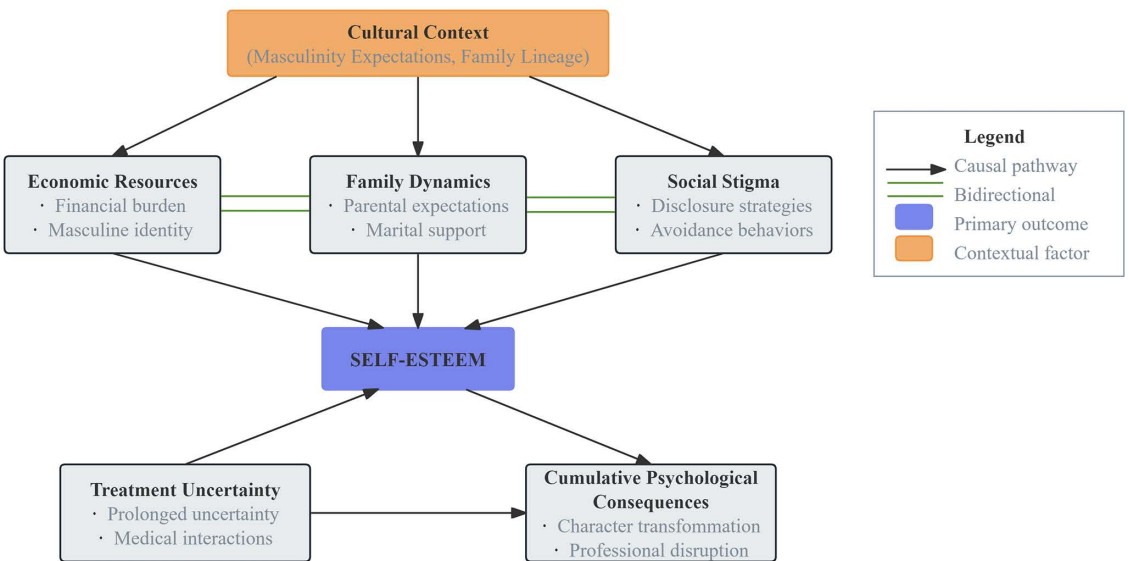

**Fig 1. Conceptual model of self-esteem in men with azoospermia.**

cumulative psychological burden, where biological infertility interacts with socioeconomic and relational contexts to erode self-worth.

While our regression model achieved statistical significance (F = 9.370, $p < 0.001$), the modest explained variance ($R^2 = 0.151$) confirms that self-esteem in this population is influenced by complex, interlocking factors beyond measured sociodemographic variables. This aligns with established psychological frameworks positing self-esteem as a multidimensional construct [26]. Supporting this perspective, Abulizi et al. demonstrated that fertility-related psychological outcomes emerge through multiple pathways, with positive thinking mediating 12–16% of effects via stress and social support mechanisms, a finding that mirrors our observed complexity [33]. The limited variance explanation quantitatively captured further validates our mixed-methods approach, as purely quantitative measures fail to reflect the nuanced lived experiences revealed through qualitative narratives.

Our results align with and expand global evidence on infertility's economic-psychological intersections. Consistent with Kiani et al. and Elliott et al. [34,35], financial strain transcended material hardship to threaten masculine identity by compromising traditional provider roles. Participants frequently conflated economic capacity with moral worth, a phenomenon amplified by reproductive failure. This resonates with Ahmadi et al.'s Middle Eastern studies [36], where infertility was perceived as fundamentally emasculating. The findings collectively exemplify Connell's hegemonic masculinity theory, wherein cultural valorization of breadwinning and biological reproduction creates psychological vulnerability when these ideals become unattainable.

The unpredictable trajectory of infertility treatments frequently induces profound psychological distress, consistent with prior reports of cyclical grief and helplessness in infertility populations [37]. Our male participants described intense anxiety surrounding treatment failures and uncertain prognoses, with prolonged emotional volatility emerging as a key contributor to self-esteem erosion. This phenomenon is particularly salient in cultural contexts like China, where Confucian values prioritize familial continuity [38]. Participants universally reported concealing their diagnosis from social networks and avoiding fertility-related discussions. These protective behaviors initially mitigated distress but ultimately reinforced isolation and internalized stigma. These findings align with Babore et al.'s work on male infertility stigma [6], while extending it by demonstrating how avoidance strategies paradoxically exacerbate long-term psychological distress through emotional suppression pathways [39].

Family relationships manifested paradoxical influences in our study. Spousal support functioned as a psychological buffer, yet intergenerational pressures, particularly from parents and in-laws, intensified feelings of inadequacy. This dichotomy reflects the complex interplay between individual and collective identities in Chinese society, where reproductive success carries multigenerational significance. As noted in prior research [37], infertile individuals often simultaneously derive comfort from marital support while experiencing profound shame when failing to meet extended family expectations. The partner correlation data further substantiate this pattern, with spousal support quality significantly moderating anxiety and depression risk [40].

Furthermore, China's Confucian cultural norms, which emphasize male responsibility in lineage continuation, significantly exacerbate feelings of guilt and shame among azoospermic patients. When unable to fulfill the socially mandated role of "continuing the family line," men experience intensified cultural pressures. This aligns with Ngai and Lam's findings that strong familial fertility expectations amplify psychological burdens and heighten sensitivity to social judgment [37]. Parallel evidence from Vietnam demonstrates how well-intentioned family interventions, particularly from in-laws, often generate counterproductive stress for infertile women, compelling conformity to reproductive expectations while worsening psychological outcomes [41]. These cross-cultural parallels underscore the necessity of culturally informed clinical interventions that address both medical and sociocultural dimensions of infertility.

While quantitative analysis revealed no significant association between treatment duration and self-esteem, qualitative narratives uncovered profound psychological impacts of prolonged treatment. This discrepancy underscores the complementary value of mixed methods in capturing complex psychosocial phenomena that elude quantitative measurement

[42]. Participants described nonlinear psychological consequences (including emotional volatility, cognitive exhaustion, and personality changes) that mirror longitudinal research on fertility treatment trajectories [43,44]. These findings suggest that treatment-related psychological effects operate through context-dependent mechanisms requiring qualitative exploration to fully elucidate.

This study's mixed methods design strengthens the validity and depth of findings through methodological triangulation, combining quantitative associations with qualitative contextualization. Nevertheless, several limitations should be acknowledged, including potential selection bias from purposive sampling and limited generalizability due to the regional focus. Additionally, we did not systematically document specific etiological factors underlying azoospermia, assess body-related variables such as body mass index, or capture the timing and context of initial diagnosis, which may independently influence self-esteem outcomes. Our findings are specific to married Chinese men, as China's regulations restrict ART access to married couples [23,24], limiting generalizability to other populations or regulatory contexts. The reliance on retrospective self-reports in qualitative interviews may introduce recall bias. Future research directions should incorporate: 1) longitudinal designs to examine temporal patterns in psychological adaptation; 2) cross-cultural comparisons to differentiate universal from context-specific impacts; and 3) dyadic approaches assessing couple dynamics. These findings highlight four key intervention priorities: implementing targeted psychosocial support programs, improving treatment affordability and access, developing anti-stigma public health campaigns, and creating family education initiatives to address intergenerational pressures.

## Conclusions

This mixed methods study elucidates the complex psychosocial consequences of azoospermia, identifying socioeconomic status, family relationships, social stigma, and treatment uncertainty as interconnected determinants of self-esteem. The findings advocate for integrated clinical interventions incorporating financial counseling to alleviate economic stressors, family-centered education programs to modify unrealistic expectations, community-based stigma reduction initiatives, and continuous mental health monitoring throughout treatment. By adopting this multidimensional care approach, healthcare systems can enhance psychological resilience and quality of life for affected men. The study underscores the necessity of moving beyond biomedical treatment paradigms to address the broader psychosocial dimensions of male infertility.

## Acknowledgments

The authors would like to thank all the participants included in this study.

## Author contributions

**Conceptualization:** Jiliang Huang, Yanshan Lin.

**Data curation:** Fangliang Zou, Hang Shi, Yu Lan.

**Formal analysis:** Jue Li, Yi Fang, Jiliang Huang, Zikai Feng, Yang Zhang, Ruiyun Chen, Yanshan Lin.

**Investigation:** Fangliang Zou, Hang Shi, Yu Lan, Yanshan Lin.

**Methodology:** Jue Li, Yanshan Lin.

**Project administration:** Jue Li, Yanshan Lin.

**Resources:** Yanshan Lin.

**Supervision:** Fangliang Zou, Jue Li, Zikai Feng, Yanshan Lin.

**Visualization:** Yi Fang, Jiliang Huang, Zikai Feng, Yang Zhang, Ruiyun Chen.

**Writing – original draft:** Fangliang Zou, Yi Fang, Jiliang Huang, Zikai Feng, Yanshan Lin.

**Writing – review & editing:** Yi Fang, Zikai Feng.

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
