## [Decision Letter · Decision Letter 0]

28 Oct 2025

Dear Dr. Lin,

We look forward to receiving your revised manuscript.

Kind regards,

Stefan Schlatt

Academic Editor

PLOS ONE

Journal Requirements:

“This research was supported by Shantou Healthcare Science and Technology Program (project ID: 240422116497013) and Teaching and Reform Research Program of Shantou University Medical College (project ID: 002-181243318).”

Additional Editor Comments:

The reviewer points to a number of shortcomings and criticize that the study population created a bias. This needs to be addressed and any shortcoming justified. I am not fully convinced that the paper can be sufficiently improved to generate full validity. However, the authors may be able by point-to-point review to fix all critical issues.

Reviewers' comments:

Reviewer's Responses to Questions

**Comments to the Author**

1. Is the manuscript technically sound, and do the data support the conclusions?

Reviewer #1: Partly

2. Has the statistical analysis been performed appropriately and rigorously?

Reviewer #1: No

3. Have the authors made all data underlying the findings in their manuscript fully available?

Reviewer #1: Yes

4. Is the manuscript presented in an intelligible fashion and written in standard English?

Reviewer #1: Yes

Reviewer #1: Dear Authors,

I've read your manuscript with interest.

There are several rreasons for concern, detailed below, which I think should be addressed in a revision of the manuscript.

1) Azoospermia is not only a consequence of testicular failure. Obstructive azoospermia and azoospermia following anabolic androgen steroids use (or abuse) are paramount examples of this. If you included men with either condition in your analysis (as inclusion and exclusion criteria do not suggest that you actually considered differential diagnosis of infertility) I would suggest repeating analysis.

2) Self-esteem of an infertile man can also depend on a series of factors that are not discussed in the paper. To be more precise: infertility might not be an issue for homosexual couples (due to the biological limitations), for unmarried men (who are missing a partner), for child-free couples (who voluntarily choose not to have children), for men who are married to a menopausal woman, and much more. I believe that your study population does not feature any of these considerations, and as such results might be biased.

3) As an infertile man myself, I would also suggest that the timing of diagnosis is another important factor. Men who find out that they are azoospermic (or, more in general, infertile) during investigations for ART are generally more "shocked" than those who were well aware of their condition since young age (e.g. Klinefelter patients).

4) following on the previous comment. Some conditions resulting in azoospermia, such as history of testicular cancer or genetic conditions, can have an independent effect on self-esteem, also mediated by other factors (e.g. body shape).

5) income per capita in table 3 is not very clear to non-Chinese readers. maybe add a note in the caption providing an esteem of the amount in $/€.

6) still in table 3, "Reproductive status" is defined as "childless" or "fertile". Given that this is a table on azoospermic men, the definition of "fertile" is dubious at least.

7) another important aspect affecting self-esteem which is completely missing from the manuscript is sexual function in the couple. Many infertile men report increased likelihood of erectile dysfunction, low libido and premature ejaculation; many women complain of hypoactive sexual desire disorder, dyspareunia and anorgasmia. These can contribute to decreased self esteem: not only his own sexual dysfunctions, but also the inability to make his partner climax can affect the psychological health of the male partner in the infertile couple.

**Do you want your identity to be public for this peer review?** For information about this choice, including consent withdrawal, please see our Privacy Policy

Reviewer #1: No

---

## [Author Response · Author response to Decision Letter 1]

20 Nov 2025

Dear Dr. Schlatt,

We would like to express our sincere gratitude to the Editorial Office and reviewers for their comments and suggestions regarding our manuscript entitled 'Self-Esteem in Crisis: Psychosocial Adaptation and Masculine Identity Among Chinese Men with Azoospermia'. Revisions have been made accordingly and our detailed responses to the comments and suggestions are provided below.

Journal Requirements:

1.Please ensure that your manuscript meets PLOS ONE's style requirements, including those for file naming.

Response: We have carefully reviewed the PLOS ONE style templates and ensured that our revised manuscript meets all formatting requirements, including file naming conventions. All files have been renamed according to the guidelines provided.

2.Please state what role the funders took in the study. If the funders had no role, please state: "The funders had no role in study design, data collection and analysis, decision to publish, or preparation of the manuscript."

Response: Thank you for your comments regarding the funding disclosure. Upon careful review, we realize that the two funding sources previously listed (Shantou Healthcare Science and Technology Program, project ID: 240422116497013; and Teaching and Reform Research Program of Shantou University Medical College, project ID: 002-181243318) are not directly related to this study. Therefore, we would like to remove these funding sources from our manuscript. The revised funding statement should read: "None." We apologize for any confusion and thank you for your attention to this matter

3.We note that you have indicated that there are restrictions to data sharing for this study. Please address data availability concerns.

Response: Thank you for raising this important concern about data availability. We have taken steps to address this issue while balancing transparency with ethical obligations:

a) Quantitative Data Availability: We have now deposited the quantitative survey data (n=216) in the Zenodo repository, which is publicly accessible at: https://doi.org/10.5281/zenodo.17641014

This dataset includes: demographic and socioeconomic variables; disease-related characteristics; Self-Esteem Scale (SES) scores; all variables used in the regression analyses.

b) Qualitative Data Restrictions: The qualitative interview transcripts and audio recordings cannot be shared publicly due to the following ethical and legal constraints:

The transcripts contain sensitive narratives about infertility experiences, sexual dysfunction, marital relationships, and psychological distress, topics that are highly stigmatized in the Chinese cultural context.

Despite pseudonymization, the detailed personal narratives risk participant identification, particularly given the small qualitative sample (n=16) and specific clinical setting.

The Institutional Review Board of the Third Affiliated Hospital of Guangzhou Medical University specifically prohibited public deposition of qualitative data when granting approval (No. 2019-066).

Participants consented to research use only, not public data sharing, in accordance with Chinese data protection regulations and hospital privacy policies.

c) Qualified Researcher Access: Researchers who meet criteria for accessing confidential qualitative data may submit requests to:

Ethics Committee: gysygcpiec@126.com; Tel: 020-81292726

Corresponding Author: Yanshan Lin (linyanshan2007@126.com)

Institution: The Third Affiliated Hospital of Guangzhou Medical University

All requests will undergo ethics committee review to ensure compliance with patient privacy protections and relevant ethical guidelines.

We have updated the Data Availability statement in the manuscript to reflect these arrangements (see Lines 550-558).

4.If the reviewer comments include a recommendation to cite specific previously published works, please review and evaluate these publications.

Response: We have carefully reviewed the reviewer's comments and note that no specific publications were recommended for citation. However, we have expanded our literature review and discussion to include additional relevant studies that strengthen the theoretical framework and contextualize our findings within the broader literature on male infertility and psychological health.

Editor's Comments:

The reviewer points to a number of shortcomings and criticizes that the study population created a bias. This needs to be addressed and any shortcoming justified.

Response: We sincerely appreciate the editor's concerns regarding potential bias in our study population. We have thoroughly addressed each of the reviewer's points below and have made substantial revisions to the manuscript to clarify our inclusion/exclusion criteria, acknowledge limitations, and justify our methodological choices. We have added detailed explanations of our study population characteristics in the Methods section and expanded the Discussion to address how these factors may have influenced our findings. We have also included a more comprehensive discussion of study limitations.

Reviewer #1:

1.Azoospermia is not only a consequence of testicular failure. Obstructive azoospermia and azoospermia following anabolic androgen steroids use (or abuse) are paramount examples of this. If you included men with either condition in your analysis (as inclusion and exclusion criteria do not suggest that you actually considered differential diagnosis of infertility) I would suggest repeating analysis.

Response: Thank you for this important comment. We would like to clarify the composition of our study population and the rationale for our inclusion criteria. We acknowledge that azoospermia encompasses distinct etiological categories, including obstructive azoospermia (OA), non-obstructive azoospermia (NOA) due to primary testicular failure, and secondary causes such as hypogonadotropic hypogonadism or medication-related suppression [1]. Our study included patients with both OA and NOA. In the Phase 1, 46.8% of OA and 53.2% of NOA patients were included, in the Phase 2, 50% of OA and 50% of NOA patients were included.

We intentionally included both groups because our research focus was on the psychosocial adaptation and self-esteem of men diagnosed with azoospermia, regardless of the underlying cause. From a psychological perspective, all azoospermic men face a common set of challenges, including the stigma associated with an infertility diagnosis, the uncertainty of treatment outcomes, significant financial burdens, and considerable pressure from family and social circles. However, we acknowledge your concern. Following your suggestion, we conducted subgroup analyses comparing OA versus NOA patients on self-esteem scores. The results showed no statistically significant difference between the two groups (t=1.67, p=0.197), supporting our integrated analytical approach (see Table 1).

2.Self-esteem of an infertile man can also depend on a series of factors that are not discussed in the paper. To be more precise: infertility might not be an issue for homosexual couples (due to the biological limitations), for unmarried men (who are missing a partner), for child-free couples (who voluntarily choose not to have children), for men who are married to a menopausal woman, and much more. I believe that your study population does not feature any of these considerations, and as such results might be biased.

Response: We appreciate the reviewer's thoughtful consideration. However, we clarify that our study population reflects China's regulatory framework rather than selection bias. In China, ART services are restricted exclusively to legally married heterosexual couples as medical treatment for diagnosed infertility, as mandated by the Ministry of Health regulations established in 2001 [2]. Patients must provide marriage certificates and birth permission certificates to access treatment, and ART clinics are prohibited from serving single individuals, unmarried couples, or same-sex partners [3]. Therefore, our study population—married heterosexual Chinese men with azoospermia, represents the entire legally eligible population that can access ART services in mainland China, not a biased subset.

We acknowledge this creates a culturally and legally specific context that may differ from Western settings. However, this specificity defines our study boundaries rather than constituting bias, accurately reflecting the population experiencing psychosocial impacts of azoospermia within China's healthcare system. We have added clarifications in the manuscript (see Lines 127-129 in Methods; Lines 506-508 in Discussion).

3.As an infertile man myself, I would also suggest that the timing of diagnosis is another important factor. Men who find out that they are azoospermic (or, more in general, infertile) during investigations for ART are generally more "shocked" than those who were well aware of their condition since young age (e.g. Klinefelter patients).

Response: Thank you for sharing this valuable perspective from personal experience. We completely agree that the timing and context of azoospermia diagnosis represents an important psychological variable. Men who discover azoospermia unexpectedly during fertility investigations may experience more acute psychological shock compared to those with long-standing awareness of their condition, such as Klinefelter syndrome patients diagnosed in adolescence.

We did collect temporal variables including infertility duration, treatment duration, and the gap time from discovering infertility to starting treatment (see Table 1). We conducted univariate analyses examining associations between these variables and self-esteem scores. While we observed trends suggesting longer infertility duration correlated with lower self-esteem, these associations did not reach statistical significance (infertility time: F=2.503, p=0.060; treatment time: F=1.931, p=0.126).

However, we acknowledge important limitations in our cross-sectional design. We did not systematically collect data on whether participants had pre-existing suspicion of fertility issues or the precise context of their initial diagnosis. Our qualitative interviews revealed variability in diagnostic experiences, with some participants describing profound shock while others described gradual awareness, but we did not systematically analyze these differences. We have added this as a limitation and suggested it as an important direction for future longitudinal research examining psychological trajectories from initial diagnosis through treatment. Thank you for this thoughtful comment that has strengthened our manuscript (see Table 1).

4.Following on the previous comment, some conditions resulting in azoospermia, such as history of testicular cancer or genetic conditions, can have an independent effect on self-esteem, also mediated by other factors (e.g. body shape).

Response: Thank you for this important comment. We would like to clarify that our study population did not include patients with history of testicular cancer or other major medical conditions. Our exclusion criteria specifically excluded patients with major comorbidities or psychiatric disorders. Additionally, as our participants were recruited from a reproductive medicine center, they represent a relatively healthy population actively seeking fertility treatment. Patients with significant underlying medical conditions that could independently affect self-esteem, such as cancer history or severe genetic syndromes with associated physical manifestations, would typically not be primary candidates for assisted reproductive treatment or would have been excluded based on our criteria.

However, we acknowledge that we did not systematically document all specific etiological factors or assess body-related variables such as body mass index that could potentially influence self-esteem. We have added this as a consideration in the limitations section. Thank you for this valuable feedback (see Lines 503-506 in Discussion).

5.Income per capita in Table 3 is not very clear to non-Chinese readers. Maybe add a note in the caption providing an estimate of the amount in $/€.

Response: We thank the reviewer for this helpful suggestion. To enhance clarity for international readers, we have added a footnote to Table 3 providing currency conversions. Based on the average 2024 exchange rates, we have converted all income brackets from Chinese Yuan (CNY) to US Dollars and Euros. The conversions are now clearly indicated in the table footnote, allowing international readers to better understand the socioeconomic context of our study participants. This revision has been implemented in the updated Table 1 and Table 3.

6.Still in Table 3, "Reproductive status" is defined as "childless" or "fertile". Given that this is a table on azoospermic men, the definition of "fertile" is dubious at least.

Response: We appreciate the reviewer raising this important terminological concern. We acknowledge that the term "fertile" is indeed misleading in the context of men diagnosed with azoospermia and creates conceptual confusion. To clarify, azoospermia can be classified as either primary (men who have never fathered children) or secondary (men who previously fathered children through natural conception but later developed azoospermia due to acquired conditions such as varicocele, testicular injury, infections, hormonal changes, or other environmental and lifestyle factors) [4]. Recent case reports have documented secondary azoospermia occurring after various medical interventions, with all affected patients having previously fathered children naturally [5].

In our study sample, participants categorized as having "reproductive status: fertile" were men who had previously achieved biological fatherhood (i.e., had existing children) but were subsequently diagnosed with azoospermia, representing cases of secondary azoospermia. The term "childless" referred to men with primary azoospermia who had never fathered biological children.

We agree that the original terminology was confusing and potentially misleading. Therefore, we have revised the "Reproductive status" variable in Table 3 by changing the category labels from "childless versus fertile" to "No children versus Has children" and added a detailed footnote explaining that "No children" represents primary azoospermia and "Has children" represents secondary azoospermia where men had previously fathered biological children before developing the condition. This revision more accurately reflects the clinical reality that all participants in our study have current azoospermia diagnoses, while appropriately distinguishing between those who have and have not previously achieved biological fatherhood. We have updated Table 3 with the revised terminology and explanatory footnote accordingly (see Table 1 and Table 3; Results section, Lines 292-296).

7.Another important aspect affecting self-esteem which is completely missing from the manuscript is sexual function in the couple. Many infertile men report increased likelihood of erectile dysfunction, low libido and premature ejaculation; many women complain of hypoactive sexual desire disorder, dyspareunia and anorgasmia. These can contribute to decreased self-esteem: not only his own sexual dysfunctions, but also the inability to make his partner climax can affect the psychological health of the male partner in the infertile couple.

Response: Thank you for this important observation regarding sexual function as a factor influencing self-esteem in infertile men. We agree that sexual dysfunction can significantly impact psychological well-being in this population.

We would like to clarify that we did consider this factor in our study design. Our exclusion criteria included patients with severe sexual dysfunction, such as erectile dysfunction, low libido, and premature ejaculation. This information was obtained through patient self-report during clinical consultations and documented in medical records. We apologize for not explicitly stating this exclusion criterion in the original manuscript and have now added it to the Methods section (see exclusion criterion 4).

By excluding men with pre-existing severe sexual dysfunction, our study was better positioned to examine the specifi

---

## [Decision Letter · Decision Letter 1]

28 Nov 2025

Self-Esteem in Crisis: Psychosocial Adaptation and Masculine Identity Among Chinese Men with Azoospermia

PONE-D-25-35878R1

Dear Dr. Lin,

We’re pleased to inform you that your manuscript has been judged scientifically suitable for publication and will be formally accepted for publication once it meets all outstanding technical requirements.

Kind regards,

Stefan Schlatt

Academic Editor

PLOS ONE

Additional Editor Comments (optional):

Reviewers' comments:

Reviewer's Responses to Questions

**Comments to the Author**

Reviewer #1: All comments have been addressed

2. Is the manuscript technically sound, and do the data support the conclusions?

Reviewer #1: Yes

3. Has the statistical analysis been performed appropriately and rigorously?

Reviewer #1: Yes

4. Have the authors made all data underlying the findings in their manuscript fully available?

Reviewer #1: Yes

5. Is the manuscript presented in an intelligible fashion and written in standard English?

Reviewer #1: Yes

Reviewer #1: Thanks for addressing my previous concerns in a detailed manner. I have no further comments for the present paper.

**Do you want your identity to be public for this peer review?** For information about this choice, including consent withdrawal, please see our Privacy Policy

Reviewer #1: **Yes:** Andrea Sansone

---

## [Editor Report · Acceptance letter]

PONE-D-25-35878R1

PLOS One

Dear Dr. Lin,

I'm pleased to inform you that your manuscript has been deemed suitable for publication in PLOS One. Congratulations! Your manuscript is now being handed over to our production team.

Kind regards,

on behalf of

Dr. Stefan Schlatt

Academic Editor

PLOS One